# Distinct Roles for RAB10 and RAB29 in Pathogenic LRRK2-Mediated Endolysosomal Trafficking Alterations

**DOI:** 10.3390/cells9071719

**Published:** 2020-07-17

**Authors:** Pilar Rivero-Ríos, Maria Romo-Lozano, Belén Fernández, Elena Fdez, Sabine Hilfiker

**Affiliations:** 1Institute of Parasitology and Biomedicine “López-Neyra”, Consejo Superior de Investigaciones Científicas (CSIC), Avda del Conocimiento s/n, 18016 Granada, Spain; mriveror@umich.edu (P.R.-R.); maria.romo@ipb.csic.es (M.R.-L.); belenfernandez@ipb.csic.es (B.F.); fdez@ipb.csic.es (E.F.); 2Life Sciences Institute, University of Michigan, Ann Arbor, MI 48109, USA; 3Department of Anesthesiology, New Jersey Medical School, Rutgers, The State University of New Jersey, Newark, NJ 07103, USA

**Keywords:** Parkinson’s disease, LRRK2, RAB10, RAB29, endolysosome, Golgi

## Abstract

**Summary Statement:**

Pathogenic LRRK2 expression causes endolysosomal trafficking alterations by impairing RAB10 function, and these alterations are rescued by RAB29 independent of its Golgi localization.

**Abstract:**

Mutations in the gene encoding leucine-rich repeat kinase 2 (LRRK2) cause familial Parkinson’s disease, and sequence variations are associated with the sporadic form of the disease. LRRK2 phosphorylates a subset of RAB proteins implicated in secretory and recycling trafficking pathways, including RAB8A and RAB10. Another RAB protein, RAB29, has been reported to recruit LRRK2 to the Golgi, where it stimulates its kinase activity. Our previous studies revealed that G2019S LRRK2 expression or knockdown of RAB8A deregulate epidermal growth factor receptor (EGFR) trafficking, with a concomitant accumulation of the receptor in a RAB4-positive recycling compartment. Here, we show that the G2019S LRRK2-mediated EGFR deficits are mimicked by knockdown of RAB10 and rescued by expression of active RAB10. By contrast, RAB29 knockdown is without effect, but expression of RAB29 also rescues the pathogenic LRRK2-mediated trafficking deficits independently of Golgi integrity. Our data suggest that G2019S LRRK2 deregulates endolysosomal trafficking by impairing the function of RAB8A and RAB10, while RAB29 positively modulates non-Golgi-related trafficking events impaired by pathogenic LRRK2.

## 1. Introduction

Mutations in the LRRK2 gene are a common cause of familial Parkinson’s disease (PD) and are also observed in sporadic PD patients, indicating a key role for LRRK2 across the entire disease spectrum [1,2,3]. LRRK2 is a large multidomain protein kinase, and the most prominent pathogenic mutation (G2019S) situated within the kinase domain increases its activity in vitro as well as in vivo, suggesting that LRRK2 inhibitors may be therapeutically beneficial for at least LRRK2-related PD [4,5,6,7,8,9,10,11].

Recent phosphoproteomic analyses identified a small subset of RAB GTPases (RAB3, RAB8, RAB10, RAB12, RAB35, and RAB43) as endogenous substrates for the LRRK2 activity [12]. RAB proteins are master regulators of all eukaryotic membrane trafficking events [13,14,15]. They are prenylated at their C-termini, and are delivered as well as extracted from membranes by GDI (RAB GDP dissociation inhibitor). At the membrane, their nucleotide-bound state is regulated by guanine nucleotide exchange factors (GEFs), which activate the RABs, and by GTPase activating proteins (GAPs), which return RABs into their inactive, GDP-bound state [16]. In their membrane- and GTP-bound forms, RABs recruit effector proteins, which bring about various events including vesicle formation, motility, and vesicle docking at the respective target membranes [13,14,15]. Differences in the nucleotide-bound state of RAB proteins are reflected by differences in the conformation of the highly conserved switch II region [17]. LRRK2 phosphorylates the RAB proteins on their switch II regions, which consequently interferes with their binding to most regulatory and effector proteins [12,18,19]. Therefore, the LRRK2-mediated phosphorylation may cause deficits in the specific membrane trafficking pathways modulated by each of these RAB proteins.

Multiple lines of evidence indicate that LRRK2 plays a role in the endolysosomal system [20], which may be related to its ability to bind and/or phosphorylate RAB proteins [12,18,21,22,23,24,25,26,27]. However, the precise underlying mechanisms remain poorly defined. In previous studies, we have analyzed the effect of pathogenic LRRK2 on the endocytic trafficking of the epidermal growth factor receptor (EGFR). In contrast to other receptors, the EGFR can either follow a purely endolysosomal degradative route or be recycled back to the cell surface, and these distinct fates can be triggered by distinct ligand concentrations [28,29]. Pathogenic LRRK2 expression was found to impair the endolysosomal degradative trafficking as well as the endocytic recycling of the EGFR, with a concomitant decrease in the levels of active, GTP-bound RAB7A, and an accumulation of the receptor in a RAB4-positive recycling compartment [30,31]. All alterations due to pathogenic LRRK2 expression were mimicked by knockdown of RAB8A and rescued by expression of active RAB8A, suggesting that they are mediated, at least in part, by RAB8A inactivation [31]. These findings support the notion that the phosphorylation of RAB8A by pathogenic LRRK2 may correlate with a loss-of-function phenotype.

RAB10 serves as a prominent LRRK2 kinase substrate in all cell lines and tissues analyzed [12,18,32,33,34,35]. It is widely expressed in peripheral tissues as well as in the brain, where it is present in all cell types [36], and localized to a tubular perinuclear endocytic recycling compartment partially overlapping with RAB8A [37,38,39,40]. RAB10 has been described to modulate a variety of membrane trafficking events including exocytic polarized targeting and endocytic recycling events [41], and can perform functions partially redundant with those of RAB8A [42,43,44,45]. However, the link between RAB10 and the endolysosomal trafficking deficits owing to pathogenic LRRK2 remains unknown.

RAB29 is contained within the PARK16 locus, which is linked to Parkinson’s disease, even though it remains unclear how variants in this locus influence disease risk [2,46,47,48,49,50,51,52]. Variants within the RAB29 and the LRRK2 locus seem to impact PD in a non-additive manner [22,49,53], suggesting that the two proteins function in a common cellular pathway. In support of this notion, pathogenic G2019S LRRK2 expression causes deficits in Golgi-lysosome-related sorting events, which are mimicked by knockdown of RAB29 and rescued by RAB29 overexpression, respectively, and transcriptome and protein analyses indicate that decreased RAB29 levels correlate with increased PD risk [22]. Other studies have reported a physical interaction between LRRK2 and RAB29, and when overexpressed, RAB29 can recruit LRRK2 to the Golgi network [23,54,55,56]. Such recruitment stimulates the LRRK2 kinase activity and causes an accumulation of phospho-RABs in the same membrane compartment, likely because the phospho-proteins are unable to be extracted by GDI [56,57]. Thus, a controversial picture has emerged whereby RAB29 either rescues the pathogenic LRRK2-mediated phenotypes [22] or serves as a membrane-bound activator for the LRRK2 kinase activity, anticipated to exacerbate pathogenic LRRK2-mediated membrane trafficking phenotypes [1,56,57].

In this study, we investigated the link between RAB10 and RAB29 and the endolysosomal trafficking deficits mediated by G2019S LRRK2. We show that expression of active RAB10 rescues, while knockdown of RAB10 mimics the trafficking deficits mediated by G2019S LRRK2. Furthermore, RAB8A and RAB10 seem to perform functionally redundant roles, at least with respect to the trafficking of the specific receptor analyzed here. In contrast, while not required for the endolysosomal trafficking of the EGFR per se, a modest increase in RAB29 levels rescues the pathogenic LRRK2-mediated deficits. Such rescue occurs independent of Golgi integrity and independent of the Golgi localization of RAB29, highlighting the possibility that this RAB protein may play additional physiological roles important for proper endolysosomal trafficking unrelated to its proposed role as a LRRK2 activator.

## 2. Materials and Methods

### 2.1. DNA Constructs and Site-Directed Mutagenesis

Triple flag-tagged LRRK2 constructs, as well as N-terminally tagged RAB8A, RAB10, RAB29, RAB7A, and RAB4 constructs, have been previously described [31,39,55,58]. The siRNA-resistant form of RAB10 was generated by introducing three silent mutations into the target sequence of the seed region of the RAB10-siRNA (Ambion, Thermo Fisher, Waltham, MA, USA, ID s21391), altering the original sequence by site-directed mutagenesis (QuickChange, Stratagene, San Diego, CA, USA) with primers 5′-GTCTATTCCTATGGTGGAAATAAATGTCGTGTTGAAGGCATCATCCGAAAAACGA-3′ and 5′-TCGTTTTTCGGATGATGCCTTCAACACGACATTTATTTCCACCATAGGAATAGAC-3′. The identity of the construct was verified by sequencing of the entire coding region. DNA was prepared from bacterial cultures grown at 37 °C using PureYield^TM^ Plasmid Midiprep System (Promega, Madison, WI, USA) according to the manufacturer’s instructions.

### 2.2. Cell Culture and Transfection

HeLa cells (ATCC; American Tissue Culture Collection) were cultured in 100 mm dishes in full medium (DMEM containing 10% fetal bovine serum, non-essential amino acids, high glucose) at 37 °C in 5% CO_2_. Confluent cells were harvested with 0.05% trypsin/0.02 mM EDTA in PBS, and subcultured at a ratio of 1:2–1:3. Cells were plated onto six-well plates and transfected the following day (70–80% confluence) using Lipofectamine 2000 (LF2000) (Invitrogen, Carlsbad, CA, USA) for 4 h in DMEM, followed by replacement with full medium. Double-transfections were performed using 2 μg of LRRK2 plasmids and 500 ng of RAB plasmids of interest. Alternatively, and where indicated, cells were transfected using Jetprime in full medium overnight according to manufacturer’s instructions using the same DNA concentrations as for transfections with LF2000. The following day, transfected cells were replated at a 1:2 ratio onto coverslips in 24-well plates, and proteins were expressed for 48 h before analysis.

### 2.3. Knockdown of RAB10, RAB8A, or RAB29 by RNAi

HeLa cells were plated into six-well plates, and transfected the following day at 70–80% confluence using 25 nM siRNA and 4 μL of jetPRIME transfection reagent (Polyplus-Transfection SA, catalog number 114-15) in 200 μL of jetPRIME buffer overnight in full medium. For knockdown experiments in the presence of GFP-tagged RAB construct expression, cells were transfected with 500 ng of the indicated RAB constructs using LF2000 for 4 h in DMEM, followed by replacement with full medium, and transfection with siRNA using jetPRIME overnight, as indicated above. In all cases, cells were passaged the next day and processed 48 h after transfection. RNAi reagents included the following: Silencer Select Negative Control Number 1 siRNA (Ambion, Thermo Fisher, catalog number 4390843), Silencer Select RAB8A (Ambion, Thermo Fisher, ID s8679), Silencer Select RAB10 (Ambion, Thermo Fisher, ID s21391), and Silencer Select RAB29 (Ambion, Thermo Fisher, ID s17082). In all cases, knockdown efficacy of the various siRNA reagents was confirmed by Western blotting with the appropriate antibodies.

### 2.4. Immunofluorescence and Laser Confocal Imaging

Where indicated, HeLa cells were incubated with 5 μg/mL brefeldin A (BFA) (Sigma Aldrich, St. Louis, MO, USA) before fixation with 4% paraformaldehyde (PFA) in PBS for 20 min at room temperature. Cells were next washed in 0.5% Triton X-100 in PBS for 3 × 5 min, followed by incubation in blocking buffer (10% goat serum, 0.5% Triton X-100 in PBS) for 1 h. Coverslips were incubated with primary antibody in blocking buffer for 1 h at room temperature, followed by washes in 0.5% Triton X-100 in PBS and incubation with secondary antibodies for 1 h. Coverslips were washed with PBS, and mounted in mounting medium with DAPI (Vector Laboratories, Burlingame, CA, USA). Primary antibodies included rabbit polyclonal anti-β-COP protein (1:200; Thermo Fisher, PA1-061) and mouse monoclonal anti-flag (1:500; Sigma, F1804), and secondary antibodies included Alexa594-conjugated goat anti-rabbit (1:2000; Invitrogen) or Alexa594-conjugated goat anti-mouse (1:2000; Invitrogen), respectively. To determine the localization of GFP-tagged RAB protein variants in the absence of antibody staining, cells were fixed as described above, but only briefly permeabilized with 0.5% Triton X-100 in PBS for 3 min, followed by washes in PBS and mounting, as described above.

Images were acquired on a Leica TCS-SP5 confocal microscope using a 63 × 1.4 numerical aperture oil UV objective (HCX PLAPO CS). Single excitation for each wavelength separately was used for all acquisitions (488 nm argon laser line and a 500–545 nm emission band pass; 543 HeNe laser line and a 556–673 nm emission band pass; 405 nm UV diode and a 422–466 nm emission band pass). The same laser settings and exposure times were employed for acquisition of individual experiments, and 10 to 15 image sections of selected areas acquired with a step size of 0.4 μm. All z-stack images were analyzed and processed using Fiji.

Live-cell fluorescence microscopy to determine colocalization of fluorescent EGF with GFP-RAB4 in the absence or presence of pathogenic LRRK2 or siRNA of RABs was performed as previously described. Briefly, cells were seeded onto 35 mm glass-bottom dishes (ibidi) 24 h after transfection, and were serum-starved overnight. The following day, medium was replaced by phenol-free, serum-free DMEM (Gibco), and cells incubated with 100 ng/mL Alexa647-EGF (Invitrogen) for 20 min at 4 °C to allow for surface binding. Excess fluorescent EGF was removed by washing cells twice in ice-cold PBS, and cells were incubated for 20 min at 37 °C to allow for internalization of bound EGF. Individual z-stage images corresponding to the cell center were acquired, and the JACoP plugin of Fiji was employed for quantification of colocalization of GFP-RAB4 with Alexa647-EGF. For this purpose, images were thresholded, and the percentage of colocalization was quantified by calculating the Mander’s coefficients (M1 for red channel, and percentage of colocalization calculated as M1 × 100) [31], with 15–20 independent cells analyzed per condition per experiment.

### 2.5. Fluorescent EGF Binding and Uptake Assays

Assays were performed essentially as described [31]. Briefly, transfected HeLa cells were plated onto coverslips the day after transfection, and serum-starved overnight. The following day, medium was replaced with fresh serum-free medium containing 100 ng/mL Alexa555-EGF (Invitrogen) at 4 °C to allow for ligand binding in the absence of internalization. In parallel experiments, control cells were washed with PBS, followed by acid stripping (0.5 M NaCl, 0.2 M acetic acid, pH 2.5, 3 min, 4 °C) to confirm that fluorescent EGF was merely surface-bound under those conditions. After binding of fluorescent EGF at 4 °C, cells were washed twice in ice-cold PBS, and then transferred to prewarmed serum-free medium to allow for uptake, and cells were fixed at the indicated times (10 min and 30 min) to quantify internalized fluorescent EGF. Fixation was performed with 4% PFA in PBS for 15 min at room temperature, cells were softly permeabilized with 0.5% Triton-X100 in PBS for 3 min, and mounted with DAPI. A minimum of 20 and up to 80 independent cells were analyzed for each condition and experiment, and quantification of the number of Alexa555-EGF structures per cell was performed by an observer blind to the conditions [31].

### 2.6. Cell Extracts and Western Blotting

HeLa cells were collected 48 h after transfection, washed in PBS, and resuspended in cell lysis buffer (1% SDS in PBS containing 1 mM PMSF, 1 mM Na_3_VO_4_, and 5 mM NaF). Lysed extracts were sonicated, boiled and centrifuged at 13,500 rpm for 10 min at 4 °C, and the protein concentration of supernatants was estimated using the BCA assay (Pierce). Where indicated, HEK293T cells were transfected as previously described [31], and extracts were prepared as described above. Proteins were resolved by SDS-PAGE and electrophoretically transferred to nitrocellulose membranes. Membranes were blocked in blocking buffer (LiCOR Biosciences, Li-COR Odyssey PBS blocking buffer, Lincoln, NE, USA, 927–40000) for 1 h at room temperature, and incubated with primary antibodies in blocking buffer overnight at 4 °C. Antibodies used for immunoblotting included rabbit polyclonal anti-GFP (1:2000; Abcam, Cambridge, UK, ab6556), mouse monoclonal anti-GAPDH (1:2000; Abcam, ab9484), mouse monoclonal anti-flag (1:500; Sigma, F1804), mouse monoclonal anti-tubulin (clone DM1A; 1:10,000; Sigma), rabbit monoclonal anti-RAB8A (1:1000; Abcam, ab188574), mouse monoclonal anti-RAB10 (1:1000; Sigma; SAB53000028), rabbit polyclonal anti-RAB7 (1:1000; Sigma, R4779), mouse monoclonal anti-RAB11 (1:1000; BD Biosciences, Madrid, Spain, 610656), rabbit monoclonal anti-RAB4 (1:1000; Abcam, ab109009), sheep polyclonal anti-RAB29 (1:250; S984D, MRC-PPU Reagents), and rabbit monoclonal anti-phosphoT73-RAB10 (1:1000; Abcam, ab230261). Secondary antibodies included goat anti-rabbit or anti-mouse IRDye 800CW, and goat anti-rabbit or anti-mouse IRDye 680CW (1:14,000), and blots were imaged via near infrared fluorescent detection using Odyssey CLx Imaging System, with quantification performed using the instrument’s Image Studio software.

Alternatively, and in all cases employing antibodies raised in sheep, membranes were blocked in blocking buffer (5% milk in 0.1% Tween-20/TBS) for 8 h, and primary antibodies diluted in blocking buffer and incubated overnight at 4 °C. Membranes were washed three times in 0.1% Tween-20/TBS, followed by incubation with HRP-conjugated rabbit anti-sheep antibodies and detection using ECL reagents (Roche Diagnostic GmbH, Barcelona, Spain). A series of timed exposures were undertaken to ensure that densitometric analyses were performed at exposures within the linear range, and the films were scanned and analysed using Quantity One (Bio-Rad, Hercules, CA, USA).

### 2.7. GST-RILP Pulldown Assays for Determination of Active RAB7

GST-RILP pulldown assays were performed as previously described in detail [31,59]. In brief, freshly purified GST-RILP protein bound to GSH-Sepharose 4B beads (GE Healthcare) was incubated with extracts from transfected HEK293T cells (one 10 cm diameter dish per assay) in pulldown buffer (20 mM HEPES, pH 7.4, 100 mM NaCl, 5 mM MgCl_2_, 1% Triton X-100, and 1 mM PMSF), and incubated overnight in a rotary wheel at 4 °C. Beads were washed twice with ice-cold pulldown buffer, and bound protein eluted with 40 μL of 1× sample buffer/β-mercaptoethanol and heating for 4 min at 95 °C prior to SDS-PAGE.

### 2.8. Statistical Analysis

All data are expressed as means ± S.E.M. One-way analysis of variance (ANOVA) with Tukey’s post hoc test was employed, and significance was set at *p* < 0.05. Significance values for all data are indicated in the figure legends, and all statistical analyses and graphs were performed using Prism software version 7.0 (GraphPad, San Diego, CA, USA).

## 3. Results

### 3.1. G2019S LRRK2-Mediated Endolysosomal Trafficking Defects are Rescued by Active RAB10 and Mimicked by Knockdown of RAB10

To determine whether RAB10 modulates the pathogenic LRRK2-mediated endolysosomal trafficking deficits, we used the EGFR trafficking assay [30,31]. Upon ligand binding using high concentrations of EGF, the EGFR is internalized by clathrin-mediated endocytosis and sorted to lysosomes for degradation [28]. The surface availability of the receptor can be determined by quantifying the binding of fluorescent EGF to cells at 4 °C, and the endocytic trafficking and degradation by quantifying the amount of endocytosed fluorescent EGF at 37 °C over time, respectively. HeLa cells were co-transfected with flag-tagged G2019S LRRK2 and either with GFP or with GFP-tagged RAB10 variants, and binding and degradation of fluorescently labelled EGF quantified (Figure 1A,B). As previously described [30,31], expression of flag-tagged G2019S LRRK2 reduced the binding of fluorescent EGF at 4 °C, and impaired the clearance/degradation of internalized fluorescent EGF upon incubation of cells at 37 °C (Figure 1A–D). GFP-tagged wildtype RAB10, or GTP-locked, constitutively active RAB10-Q68L were both localized to a tubular perinuclear compartment, while GDP-locked inactive RAB10-T23N was largely cytosolic (Appendix A). Both wildtype RAB10 and RAB10-Q68L were expressed to similar degrees, and did not interfere with the co-expression of G2019S LRRK2 (Appendix A). Expression of GFP-tagged RAB10 variants on their own was without effect on EGF binding or degradation (Figure 1E,F). However, when co-expressed with pathogenic G2019S LRRK2, active RAB10-Q68L fully rescued the decrease in EGF binding and the impairment in EGFR degradation, which was not observed with wildtype RAB10 or with the inactive RAB10 variant (Figure 1C,D), suggesting that pathogenic LRRK2 may cause the inactivation of RAB10.

As another means to analyze the effect of RAB10 inactivation on EGF binding and EGFR trafficking, we performed siRNA experiments. Knockdown of RAB10 caused a pronounced decrease in RAB10 protein levels 48 h post-transfection (Figure 2A,B), while the steady-state levels of several other RAB proteins including RAB8A (Figure 2A,B) or RAB7A remained unchanged (Appendix A). The knockdown of RAB10 was accompanied by a significant decrease in EGF surface binding and EGFR degradation (Figure 2C,D). A siRNA-resistant version of RAB10, but not wildtype siRNA-sensitive RAB10, rescued the effect of RAB10 knockdown on EGF binding and EGFR trafficking (Figure 2E–G), indicating that the effects were owing to the specific knockdown of RAB10. Thus, siRNA of RAB10 mimics the endolysosomal trafficking deficits mediated by G2019S LRRK2 expression.

### 3.2. Knockdown of RAB10 Causes a Decrease in RAB7 Activity and Mistargeting of EGF into an RAB4 Compartment

We previously reported that knockdown of RAB8A mimicked the effects of G2019S LRRK2 on endolysosomal trafficking by decreasing the activity of RAB7 [31], a crucial regulator of endolysosomal trafficking pathways [60,61]. The decrease in RAB7 activity upon either G2019S LRRK2 expression or knockdown of RAB8A was associated with the accumulation of EGF in a RAB4-positive endocytic compartment, and all trafficking deficits were rescued upon expression of active RAB7A [31]. Similarly, the deficits in EGF surface binding and EGFR degradation induced upon siRNA of RAB10 were rescued when overexpressing active, GTP-locked RAB7A (RAB7A-Q67L), but not wildtype or GDP-locked RAB7A (RAB7A-T22N) (Figure 3A,B), even though all RAB7A versions were expressed to comparable degrees (Figure 3C). In addition, knockdown of RAB10 was associated with the redistribution of EGF into a vesicular recycling compartment colocalizing with RAB4, which was rescued upon active RAB7A expression (Appendix A).

To gain direct evidence for a decrease in RAB7A activity upon siRNA of RAB10, we performed effector pulldown assays. Cells were treated with either control siRNA or with RAB10-siRNA, and cell lysates were subjected to pulldowns with the RAB7-binding domain of RILP to selectively isolate active, GTP-bound RAB7 [31,59]. Studies of this type further indicate that the fraction of endogenous active RAB7 was significantly reduced upon RAB10-siRNA (Figure 3D,E).

Because either knockdown of RAB8A or of RAB10 phenocopied the effects of pathogenic G2019S LRRK2 expression, we wondered whether these RAB proteins may act in a functionally redundant manner with respect to EGF trafficking and EGFR degradation. Indeed, the decrease in EGF binding and EGFR degradation mediated by knockdown of RAB10 was rescued upon expression of active, but not wildtype or inactive RAB8A (Figure 4A,B). Conversely, the deficits in EGF binding and EGFR degradation upon siRNA of RAB8A were rescued upon expression of active, but not wildtype or inactive RAB10 (Figure 4C,D). Thus, impairing the function of either RAB8A or RAB10 causes endolysosomal trafficking deficits identical to G2019S LRRK2 expression, which are associated with a decrease in RAB7 activity and culminate in the accumulation of EGF in a non-degradative, RAB4-positive recycling compartment.

### 3.3. G2019S LRRK2-Mediated Endolysosomal Trafficking Defects are Rescued by RAB29 Expression

Previous studies indicate that overexpression of RAB29 causes recruitment of LRRK2 to the trans-Golgi network (TGN), which in turn activates LRRK2 as assessed by RAB10 phosphorylation [23,54,55,56]. Surprisingly, while GFP-tagged RAB29 largely colocalized with a Golgi marker, coexpression of RAB29 with flag-tagged G2019S LRRK2 revealed that this did not result in the efficient recruitment of LRRK2 to the Golgi complex (Figure 5A). We next assessed the effect of RAB29 expression on the pathogenic LRRK2-mediated endolysosomal trafficking deficits. When coexpressed with G2019S LRRK2, RAB29 rescued the LRRK2-mediated decrease in EGF binding and the impairment in EGFR degradation (Figure 5B,C). In contrast, two mutant RAB29 versions previously described to cause RAB29 inactivation [23,54,55] did not rescue the LRRK2-mediated endolysosomal trafficking deficits (Figure 5B,C). Furthermore, expression of the various RAB29 constructs on their own was without effect (Figure 5D,E), even though the RAB29 variants were expressed to similar degrees and did not change expression levels of G2019S LRRK2 (Figure 5F). Finally, wildtype, but not mutant RAB29 versions were also found to rescue the G2019S LRRK2-mediated accumulation of EGF in an RAB4-positive endocytic compartment (Appendix A).

We reasoned that the previously reported RAB29-mediated recruitment and activation of LRRK2 may be dependent on overexpression levels. Indeed, when employing another transfection reagent to achieve higher expression levels, GFP-RAB29 expression caused a pronounced recruitment of G2019S LRRK2 (Appendix A). Under these conditions, while expression of RAB29 variants on their own still was without effect (Appendix A), co-expression of wildtype RAB29 with G2019S LRRK2 failed to rescue the LRRK2-mediated deficits in EGF binding and EGFR degradation (Appendix A), in contrast to what was observed with low-level RAB29 expression in parallel experiments (Appendix A). Thus, GFP-RAB29 is largely localized to the Golgi complex independent of expression levels. However, and at least as analyzed here, the RAB29-mediated recruitment of LRRK2 to the Golgi complex is only evident upon higher expression levels, and is associated with an impairment of the RAB29-mediated rescue of the endolysosomal trafficking deficits owing to G2019S LRRK2 expression.

### 3.4. RAB29 Expression Rescues the Endolysosomal Trafficking Deficits Mediated by G2019S LRRK2 or Knockdown of Either RAB8A or RAB10

We next assessed whether the LRRK2-mediated endolysosomal trafficking deficits may be owing to RAB29 inactivation. Knockdown of RAB29 was without effect on EGF binding and EGFR degradation (Appendix A), in contrast to what we observed with knockdown of RAB8A [31] or RAB10 (Figure 2). Furthermore, transient disruption of the Golgi complex by brefeldin A (BFA) treatment did not alter EGF binding or EGFR trafficking per se (Appendix A), suggesting that the LRRK2-mediated endolysosomal trafficking deficits are neither mimicked by RAB29 inactivation nor dependent on Golgi integrity. Importantly, while BFA treatment caused the perinuclear dispersal of a Golgi marker as well as of GFP-RAB29 (Figure 6A,B), it did not interfere with the RAB29-mediated rescue of the deficits in EGF binding and EGFR degradation owing to the presence of G2019S LRRK2 (Figure 6C,D). Thus, the RAB29-mediated rescue of the trafficking deficits owing to G2019S LRRK2 expression does not require an intact Golgi complex.

We reasoned that RAB29 may play additional roles in membrane trafficking apart from those described at the Golgi complex, possibly overlapping with those of either RAB8A and/or RAB10. Indeed, expression of wildtype, but not inactive RAB29 variants rescued the deficits in EGF binding and EGFR degradation induced upon knockdown of either RAB10 (Figure 7A–C) or RAB8A (Figure 7D–F), with the RAB29 variants expressed to similar degrees (Figure 7C,F). Expression of wildtype RAB29 also reverted the accumulation of EGF in a RAB4-positive endocytic compartment induced upon knockdown of either RAB10 or RAB8A (Appendix A). Finally, the trafficking deficits mediated by expression of a dominant-negative RAB7A mutant were rescued by RAB29 expression (Appendix A), as previously described for active RAB8A [31]. These data indicate that RAB29 can rescue the endolysosomal trafficking deficits induced by either G2019S LRRK2 expression or by knockdown of RAB8A or RAB10 in a manner independent of its localization/function at the Golgi complex.

## 4. Discussion

In the present study, we find that the endolysosomal deficits mediated by pathogenic LRRK2 are mimicked by knockdown of RAB10 and rescued by expression of active RAB10. The deficits in endolysosomal trafficking/degradation of the EGFR by RAB10 inactivation are accompanied by the accumulation of the receptor in a RAB4-positive recycling compartment, and by a decrease in the levels of active, GTP-bound RAB7A, identical to what we previously described for pathogenic LRRK2 expression or knockdown of RAB8A [30,31]. In addition, the endolysosomal trafficking deficits mediated by knockdown of RAB10 are rescued by expression of active RAB8A and vice versa. These data are consistent with a model whereby the LRRK2-mediated deficits in endolysosomal trafficking are the result of a loss-of-function phenotype of either RAB10 or RAB8A, likely triggered by the phosphorylation and concomitant inactivation of these RAB proteins. Furthermore, our data indicate that RAB8A and RAB10 play functionally redundant roles, at least for the endocytic trafficking/degradation of the EGFR, as analyzed here. This is consistent with their colocalization to a perinuclear tubular endocytic recycling compartment [37,38,39,40], and with previous reports showing that RAB8 and RAB10 can cooperatively regulate other vesicular trafficking pathways including basolateral sorting, neurite outgrowth, and ciliogenesis [42,43,44,45]. Thus, the sum of phospho-RAB8 and phospho-RAB10 may be driving the pathogenic LRRK2-mediated vesicular trafficking deficits, and in the future, it will be important to determine the combined levels of both phospho-proteins, and whether LRRK2-mediated endolysosomal trafficking alterations occur in a manner dependent on RAB8A and RAB10 in dopaminergic neurons or other disease-relevant cell types. In addition, endocytic trafficking deficits have also been described for the dopamine D1 receptor and the transferrin receptor [62,63], and it will be interesting to determine whether they are owing to the same LRRK2-mediated inactivation of RAB8A and RAB10 as described here.

The accumulation of EGF in a RAB4-positive recycling compartment upon pathogenic LRRK2 expression or RNAi of either RAB8A or RAB10 reflects a crosstalk between recycling and degradative trafficking pathways, which has been described before [64,65,66]. For example, expression of a dominant-negative RAB4 construct alters both endocytic recyling and degradation events [64], and interfering with the integrity/function of the RAB8-positive early recycling compartment causes impaired degradation of the EGFR, which consequently accumulates in a RAB4-positive recycling compartment [65]. Thus, impairing the proper functioning of the RAB8/RAB10-positive perinuclear endocytic recycling compartment is associated with deficits in the complex crosstalk between recycling and degradative vesicular trafficking pathways, with various and potentially disease-relevant cellular outcomes.

The trafficking alterations mediated by pathogenic LRRK2 or knockdown of either RAB8A or RAB10 all converge on decreasing the amounts of GTP-bound, active RAB7. RAB7A plays a key role in the endolysosomal system by regulating trafficking from the late endosome/multivesicular body to the lysosome, and is also implicated in the process of lysosome reformation [60,61,67,68]. RAB7A does not serve as a substrate for the LRRK2 kinase activity in vitro or in intact cells [12,18,31], indicating that the decrease in RAB7A activity mediated by pathogenic LRRK2 expression occurs by indirect means. The specificity of intracellular vesicular trafficking pathways is often ensured by a so-called RAB cascade, whereby the effector of the upstream RAB protein acts as an activator (GEF) for the downstream RAB protein [69,70,71,72]. Such a RAB cascade has been described for the trafficking between RAB5-positive early endosomes and RAB7-positive late endosomes, whereby RAB5 is necessary and sufficient to drive GEF-dependent activation of RAB7 [73]. In the future, it will be important to determine whether a similar RAB cascade operates between RAB8/RAB10-positive and RAB7-positive compartments, which may underlie the observed decrease in RAB7 activity due to RAB8/RAB10 phosphorylation/inactivation.

In contrast to RAB8 and RAB10, knockdown of RAB29 did not cause endolysosomal trafficking deficits of the EGFR, suggesting that, under normal conditions, RAB29 is not required for this process. However, knockdown or knockout of RAB29 causes deficits in retrograde sorting events between the late endosome and the Golgi complex, which is crucial for the correct trafficking of lysosomal enzymes and associated with defects in lysosomal homeostasis [22,27,74]. Deficits in the trafficking/degradation of the EGFR may not have been evident owing to the incomplete and/or transient nature of the RAB29 knockdown, because alterations in the correct processing of lysosomal hydrolases have only been described in RAB29 knockout contexts, and may thus only arise as a consequence of the prolonged missorting of cargo [27,74].

While knockdown of RAB29 was without effect on EGFR trafficking, RAB29 expression rescued the pathogenic LRRK2-mediated endolysosomal trafficking deficits. This finding is consistent with previous studies reporting a beneficial effect of increasing RAB29 protein levels on Golgi-lysosome sorting events, neurite outgrowth, and dopaminergic cell survival in Drosophila neurons, and with transcriptome analysis suggesting that RAB29 risk variants correlate with a decrease in RAB29 expression levels [22]. As other studies have suggested that increased PD risk may be associated with increased RAB29 levels [23], further studies using newly generated antibodies specific against RAB29 are warranted to correlate PD risk with possible alterations in RAB29 protein levels.

The rescue of the LRRK2-mediated deficits upon RAB29 expression occurred in a manner largely independent of Golgi integrity, suggesting that RAB29 may play role(s) in membrane trafficking in addition to those described at the Golgi complex. Indeed, previous studies indicate that RAB29 colocalizes and interacts with RAB8, and regulates receptor recycling in T cells as well as the trafficking of endosome-derived ciliary cargo at the base of the cilium [75]. As RAB29 expression can functionally complement for the knockdown of either RAB8A or RAB10, it is tempting to speculate that at least a fraction of RAB29 may be localized to the recycling compartment, where it may be able to perform functions overlapping with those of RAB8A and/or RAB10 (Figure 8). Finally, our data show that the reported recruitment of LRRK2 to the Golgi complex by RAB29 [23,54,55,56] is only observed when RAB29 is overexpressed to a larger extent. This is consistent with findings indicating that most endogenous LRRK2 is not present on the Golgi complex, and that phospho-RAB8A/RAB10 is not trapped on the Golgi complex under normal conditions either [39,55,57]. While further work is required to understand the link between RAB29, LRRK2, and the etiology of PD, the present data highlight the possibility that, under more physiological conditions, RAB29 may be a positive regulator of non-Golgi-related membrane trafficking events impaired by pathogenic LRRK2, rather than controlling the localization and activation of LRRK2 at the Golgi complex.

## Figures and Tables

**Figure 1 cells-09-01719-f001:**
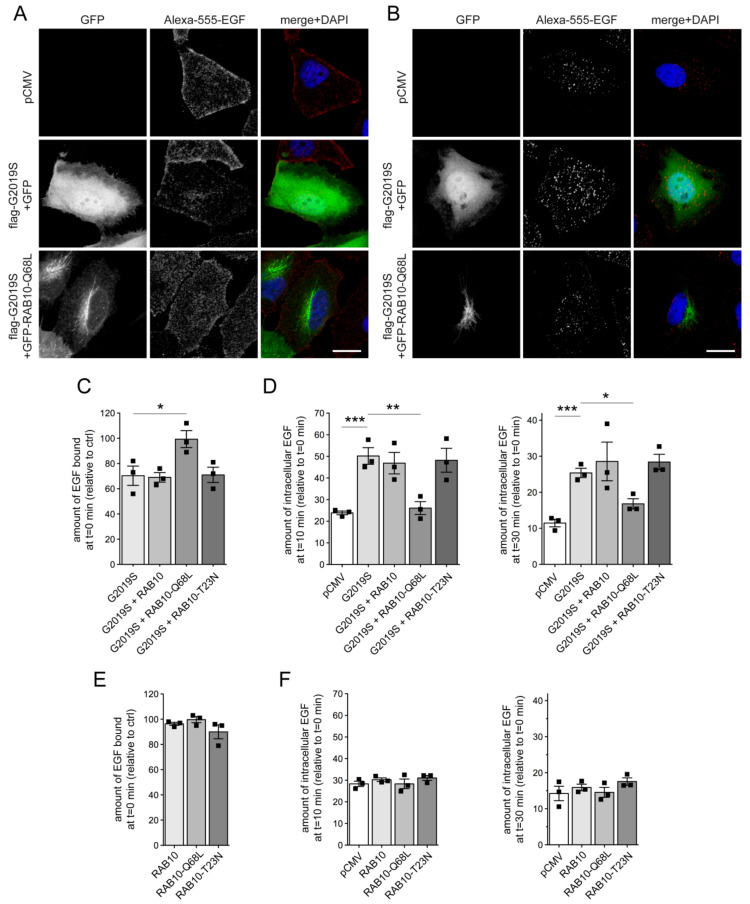
Active RAB10 rescues the G2019S leucine-rich repeat kinase 2 (LRRK2)-mediated deficit in epidermal growth factor (EGF) binding and degradation. (**A**) HeLa cells were transfected with either pCMV, or cotransfected with flag-tagged G2019S LRRK2 and GFP or GFP-tagged RAB10-Q68L as indicated. Cells were incubated with Alexa555-EGF for 20 min at 4 °C, followed by washing to remove unbound fluorescent EGF before fixation (t = 0 min). Scale bar, 10 μm. (**B**) Same as in (**A**), but upon incubation and washing, cells were shifted to 37 °C for 10 min to allow for the internalization and degradation of fluorescent EGF. Scale bar, 10 μm. (**C**) Cells were co-transfected with G2019S LRRK2 and either GFP, or GFP-tagged RAB10 constructs as indicated, and the amount of surface-bound fluorescent EGF was quantified. N = 3 experiments; * *p* < 0.05. (**D**) Cells were co-transfected as indicated, and the amount of internalized Alexa555-EGF in transfected cells was quantified after 10 min (left) and 30 min (right) of internalization, with values normalized to the amount of fluorescent EGF binding at t = 0. N = 3 experiments; * *p* < 0.05; ** *p* < 0.01; *** *p* < 0.005. (**E**) The amount of surface-bound fluorescent EGF was quantified at t = 0 min from cells transfected with the indicated GFP-tagged RAB10 constructs, and normalized to EGF surface binding of pCMV-transfected cells (ctrl). N = 3 experiments. (**F**) The amount of fluorescent EGF was quantified after 10 min (left) and 30 min (right) upon internalization, and normalized to the amount of Alexa555-EGF binding for each condition at t = 0 min, thus reflecting the percentage of internalized bound fluorescent EGF. N = 3 experiments. All bars represent mean ± s.e.m.

**Figure 2 cells-09-01719-f002:**
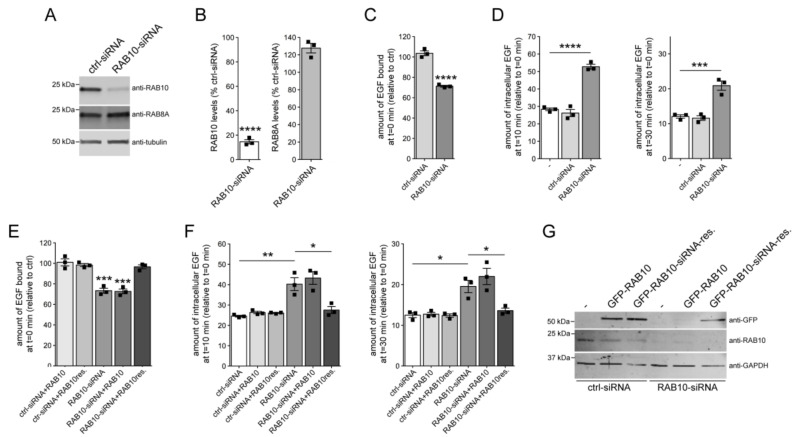
Knockdown of RAB10 mimics the EGF trafficking deficits observed upon G2019S LRRK2 expression. (**A**) HeLa cells were either transfected with ctrl-siRNA or RAB10-siRNA, and cell extracts were analyzed by Western blotting for RAB10 protein levels, RAB8A protein levels, or tubulin as loading control. (**B**) Quantification of RAB10 and RAB8A protein levels in the presence of RAB10-siRNA, normalized to the levels in the presence of ctrl-siRNA. Bars represent mean ± s.e.m. (N = 3 independent experiments; **** *p* < 0.001). (**C**) Cells were either left untreated (-), or treated with ctrl-siRNA or RAB10-siRNA, and the amount of surface-bound fluorescent EGF was quantified. N = 3 independent experiments; **** *p* < 0.001. (**D**) Cells were either untreated (-), or treated with ctrl-siRNA or RAB10-siRNA, and internalized fluorescent EGF was quantified at 10 min (left) and 30 min (right). N = 3 independent experiments; *** *p* < 0.005; **** *p* < 0.001. (**E**) Cells were either transfected with ctrl-siRNA or RAB10-siRNA in the absence or presence of wildtype or siRNA-resistant GFP-tagged RAB10 as indicated, and the amount of surface-bound fluorescent EGF was quantified. N = 3 independent experiments. *** *p* < 0.005. (**F**) Cells were either transfected with ctrl-siRNA or RAB10-siRNA in the absence or presence of wildtype or siRNA-resistant GFP-RAB10, and internalized fluorescent Alexa555-EGF quantified upon 10 min (left) or 30 min (right) of internalization. N = 3 independent experiments. * *p* < 0.05; ** *p* < 0.01. (**G**) Cells were either transfected with ctrl-siRNA or RAB10-siRNA, and with either wildtype or siRNA-resistant GFP-tagged RAB10 as indicated, and cell extracts (30 μg) analyzed by Western blotting for GFP-RAB10 levels, endogenous RAB10 levels, and GAPDH as loading control.

**Figure 3 cells-09-01719-f003:**
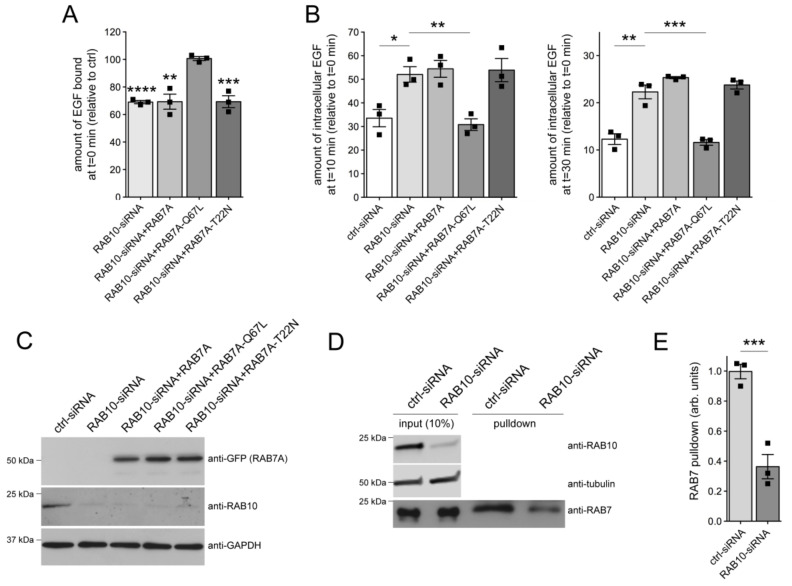
Knockdown of RAB10 decreases RAB7 activity. (**A**) HeLa cells were either transfected with ctrl-siRNA, or with RAB10-siRNA in the absence or presence of GFP-tagged RAB7A constructs as indicated, and surface-bound fluorescent EGF was quantified. N = 3 independent experiments. ** *p* < 0.01; *** *p* < 0.005; **** *p* < 0.001. (**B**) Cells were either transfected with ctrl-siRNA, or with RAB10-siRNA in the absence or presence of GFP-tagged RAB7A constructs as indicated, and internalized fluorescent EGF was quantified at 10 min (left) and 30 min (right) upon internalization. N = 3 independent experiments. * *p* < 0.05; ** *p* < 0.01; *** *p* < 0.005. (**C**) Cells were treated with ctrl-siRNA or RAB10-siRNA, and transfected with the indicated RAB7A constructs, and cell extracts (30 μg) were analyzed by Western blotting for GFP-RAB7A protein levels, endogenous RAB10 protein levels, and GAPDH as loading control. (**D**) Cells were either treated with ctrl-siRNA or RAB10-siRNA. The RAB7-binding domain of RILP coupled to GST was used to pull down the GTP-bound form of RAB7 from cell lysates (300 μg), and 10% of input was run alongside pulldowns to show equal levels of total RAB7 protein in ctrl-siRNA or RAB10-siRNA-treated cells. The levels of RAB10 and tubulin were analyzed on a separate gel. (**E**) Quantification of the type of experiments is depicted in (**D**), with the amount of RAB7 isolated by GST-RILP expressed relative to input. N = 3 independent experiments. *** *p* < 0.005.

**Figure 4 cells-09-01719-f004:**
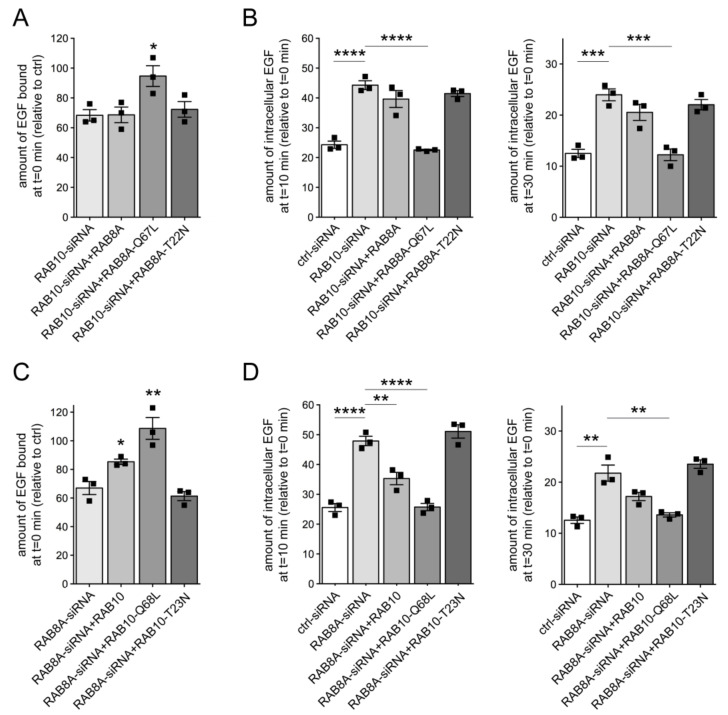
RAB8A and RAB10 are functionally redundant in regulating endolysosomal EGF trafficking. (**A**) HeLa cells were either transfected with ctrl-siRNA or RAB10-siRNA, and co-transfected with GFP-tagged RAB8A constructs as indicated, and the amount of surface-bound fluorescent EGF was quantified. N = 3 independent experiments; * *p* < 0.05. (**B**) Cells were transfected with ctrl-siRNA or RAB10-siRNA, and co-transfected with GFP-tagged RAB8A constructs as indicated, and internalized fluorescent EGF was quantified at 10 min (left) and 30 min (right). N = 3 independent experiments; *** *p* < 0.005; **** *p* < 0.001. (**C**) HeLa cells were either transfected with ctrl-siRNA or RAB8A-siRNA, and co-transfected with GFP-tagged RAB10 constructs as indicated, and the amount of surface-bound fluorescent EGF was quantified. N = 3 independent experiments; * *p* < 0.05; ** *p* < 0.01. (**D**) Cells were transfected with ctrl-siRNA or RAB8A-siRNA, and co-transfected with GFP-tagged RAB10 constructs as indicated, and internalized fluorescent EGF was quantified at 10 min (left) and 30 min (right). N = 3 independent experiments; ** *p* < 0.01; **** *p* < 0.001.

**Figure 5 cells-09-01719-f005:**
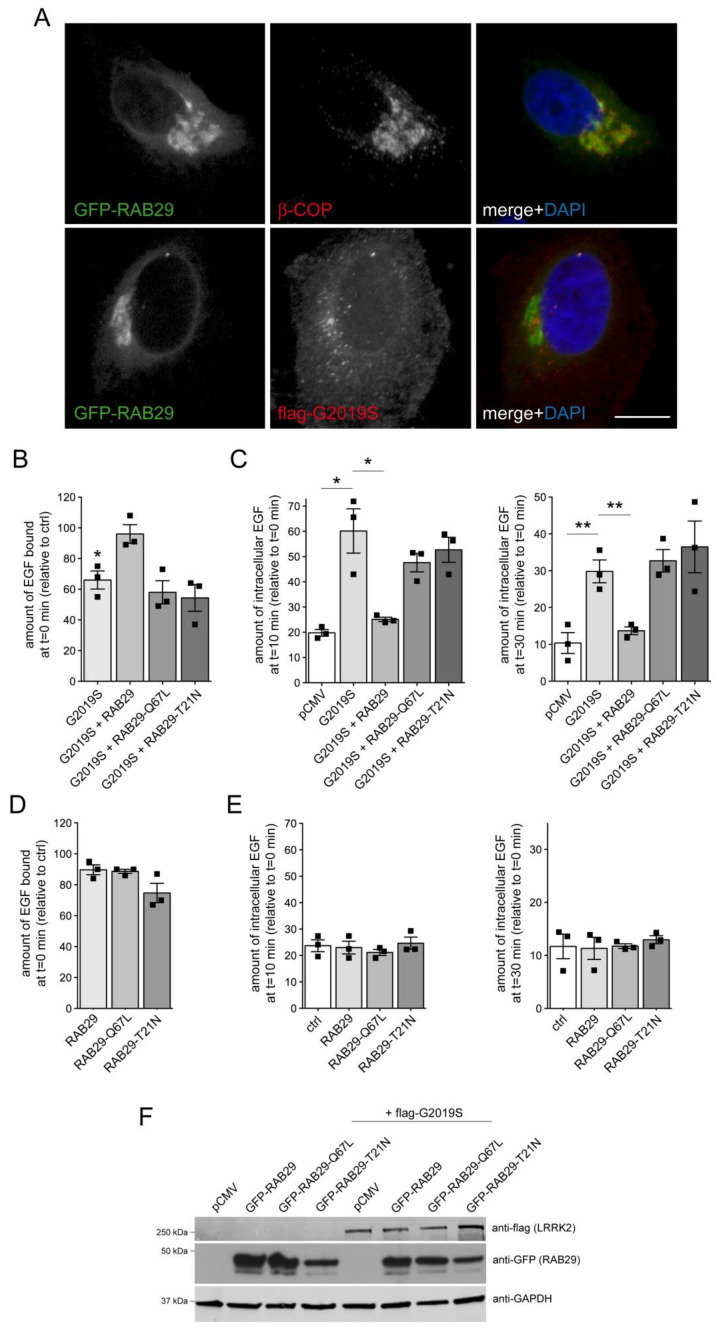
RAB29 rescues the G2019S LRRK2-mediated EGF trafficking deficits. (**A**) Top: Example of HeLa cell transfected with GFP-tagged RAB29 using Lipofectamine 2000 (LF2000), and stained with Golgi marker (β-COP) and DAPI. Bottom: Example of HeLa cell co-transfected with GFP-tagged RAB29 and flag-tagged G2019S LRRK2 using LF2000, and stained with flag antibody and DAPI. Scale bar, 10 μm. (**B**) Cells were transfected with either pCMV, or cotransfected with flag-tagged G2019S LRRK2 and GFP or GFP-tagged RAB29 constructs as indicated, and surface-bound fluorescent EGF quantified. N = 3 independent experiments. * *p* < 0.05. (**C**) Cells were transfected with either pCMV, or cotransfected with flag-tagged G2019S LRRK2 and GFP or GFP-tagged RAB29 constructs, and internalized fluorescent EGF was quantified at 10 min (left) and 30 min (right) upon internalization. N = 3 independent experiments. * *p* < 0.05; ** *p* < 0.01. (**D**) HeLa cells were transfected with the indicated GFP-tagged RAB29 constructs using Lipofectamine 2000 (LF2000), and surface-bound fluorescent EGF quantified. N = 3 independent experiments. (**E**) Cells were transfected with the indicated GFP-tagged RAB29 constructs, and internalized fluorescent EGF was quantified at 10 min (left) and 30 min (right) upon internalization. N = 3 independent experiments. (**F**) HeLa cells were transfected with empty pCMV vector, with GFP-tagged RAB29 constructs, or with flag-G2019S LRRK2 along with GFP-tagged RAB29 constructs as indicated, and cell extracts (30 μg) analyzed by Western blotting for flag, GFP, and GAPDH as loading control.

**Figure 6 cells-09-01719-f006:**
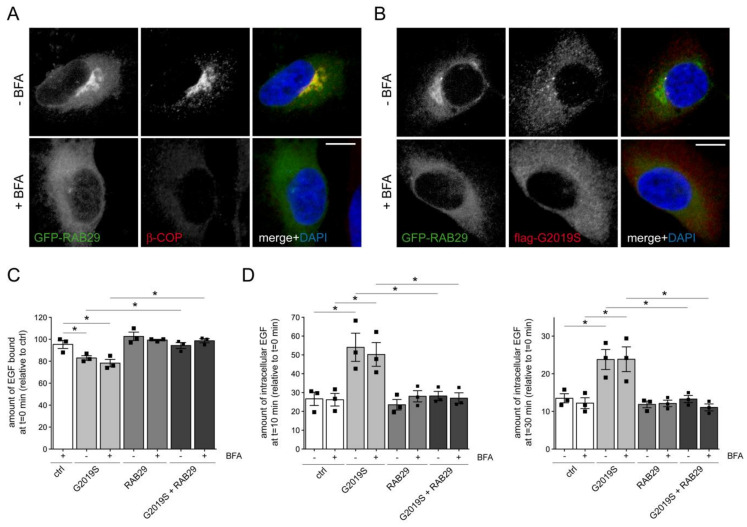
The G2019S LRRK2-mediated deficit in EGF trafficking is rescued by RAB29 independent of Golgi integrity. (**A**) Example of HeLa cell transfected with GFP-RAB29 using Lipofectamine 2000 (LF2000), and stained for β-COP and DAPI in either the absence or presence of BFA treatment (5 μg/mL, 2 h) as indicated. Scale bar, 10 μm. (**B**) Example of HeLa cell cotransfected with GFP-RAB29 and flag-tagged G2019S LRRK2 using LF2000, and stained for flag and DAPI in either the presence or absence of BFA treatment (5 μg/mL, 2 h) as indicated. Scale bar, 10 μm. (**C**) HeLa cells were either transfected with pCMV (ctrl), flag-tagged G2019S LRRK2, GFP-tagged RAB29, or cotransfected with flag-tagged G2019S LRRK2 and GFP-tagged RAB29 using LF2000 as indicated, and either treated with or without BFA (5 μg/mL, 2 h) before determination of the amount of surface-bound fluorescent EGF. N = 3 independent experiments; * *p* < 0.05. (**D**) HeLa cells were either left untreated (ctrl), transfected with flag-tagged G2019S LRRK2, GFP-tagged RAB29, or cotransfected with flag-tagged G2019S LRRK2 and GFP-tagged RAB29 using LF2000, either treated with or without BFA (5 μg/mL, 2 h) as indicated, and internalized fluorescent EGF was quantified at 10 min (left) and 30 min (right) upon internalization. N = 3 independent experiments. * *p* < 0.05.

**Figure 7 cells-09-01719-f007:**
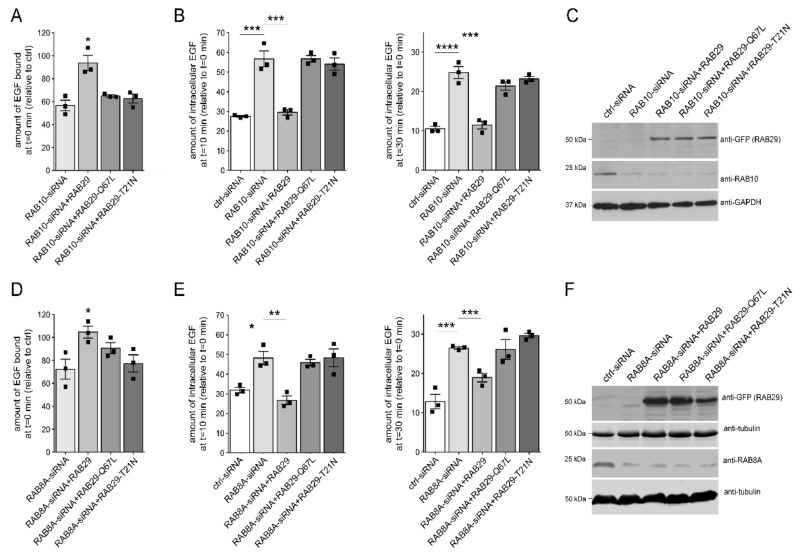
Deficits in EGF trafficking due to knockdown of RAB8A or RAB10 are rescued upon RAB29 expression. (**A**) HeLa cells were either transfected with ctrl-siRNA or RAB10-siRNA, and cotransfected with GFP-tagged RAB29 constructs as indicated, and the amount of surface-bound fluorescent EGF quantified. N = 3 independent experiments; * *p* < 0.05. (**B**) Cells were transfected with ctrl-siRNA or RAB10-siRNA, and cotransfected with GFP-tagged RAB29 constructs as indicated, and internalized fluorescent EGF was quantified at 10 min (left) and 30 min (right). N = 3 independent experiments; *** *p* < 0.005; **** *p* < 0.001. (**C**) Cells were either transfected with ctrl-siRNA or RAB10-siRNA, and transfected with GFP-tagged RAB29 constructs as indicated, and cell extracts (30 μg) were analyzed by Western blotting for GFP-RAB29 levels, endogenous RAB10 levels, and GAPDH as loading control. (**D**) HeLa cells were either transfected with ctrl-siRNA or RAB8A-siRNA, and cotransfected with GFP-tagged RAB29 constructs as indicated, and the amount of surface-bound fluorescent EGF was quantified. N = 3 independent experiments; * *p* < 0.05. (**E**) Cells were transfected with ctrl-siRNA or RAB8A-siRNA, and cotransfected with GFP-tagged RAB29 constructs as indicated, and internalized fluorescent EGF was quantified at 10 min (left) and 30 min (right). N = 3 independent experiments; * *p* < 0.05; ** *p* < 0.01; *** *p* < 0.005. (**F**) Cells were either transfected with ctrl-siRNA or RAB8A-siRNA, and transfected with GFP-tagged RAB29 constructs as indicated, and cell extracts (30 μg) were analyzed by Western blotting for GFP-RAB29 levels, endogenous RAB8A levels, and tubulin as loading control.

**Figure 8 cells-09-01719-f008:**
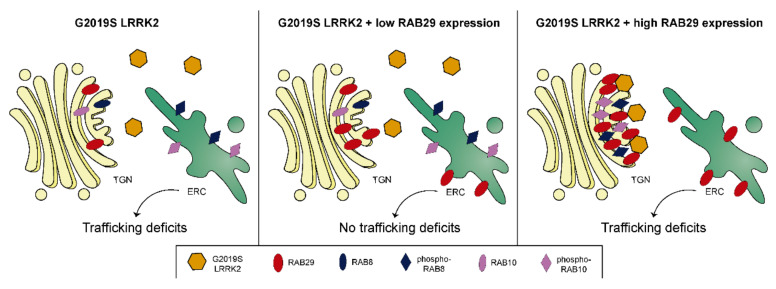
Model for the RAB29-mediated regulation of G2019S LRRK2 localization and rescue of trafficking deficits. Schematics summarizing the RAB29-mediated modulation of trafficking defects owing to pathogenic LRRK2. Left: RAB8A and RAB10 are mainly localized to the early recycling compartment (ERC), while endogenous RAB29 is mainly localized to the Golgi complex. Pathogenic G2019S LRRK2 causes phosphorylation (and thus inactivation) of RAB8A/RAB10 at the ERC, resulting in deficits of the endolysosomal trafficking of the EGFR (trafficking deficits). Middle: upon low/moderate overexpression of RAB29, G2019S LRRK2 remains cytosolic, while RAB29 localizes to the Golgi (TGN) and possibly also the ERC. G2019S LRRK2 causes phosphorylation/inactivation of RAB8A/RAB10 at the ERC, but non-Golgi-localized RAB29 is able to rescue the pathogenic LRRK2-mediated trafficking deficits, perhaps by positively regulating the remaining non-phosphorylated RAB8A/RAB10 at the ERC. Right: upon high overexpression of RAB29, G2019S LRRK2 is recruited to the Golgi complex where it phosphorylates and “traps” RAB8A/RAB10, resulting in a depletion of non-phosphorylated RAB8A/10 at the ERC. Under these conditions, non-Golgi-localized RAB29 is no longer able to rescue the pathogenic LRRK2-mediated trafficking deficits.

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
