# Peer review of "Distinct Roles for RAB10 and RAB29 in Pathogenic LRRK2-Mediated Endolysosomal Trafficking Alterations"

_cells, 2020, doi:10.3390/cells9071719_

Round 1

Reviewer 1 Report

This manuscript focused on a subset of RAB proteins that are linked with a LRRK2 mutation observed in Parkinson's disease patients. This is follow-up paper to their recent publications looking at the role RAB8A and Rab29 play in the cell under basal conditions and with those transfected with G2019S LRRK2 mutation. Overall, the experiments were conducted with all the proper controls and biological replicates using appropriate statistical analyses. I feel the authors performed adequate experiments to answer the biological questions. The authors show for the first time that the G2019S LRRK2 mutation impairs endolysosomal trafficking/degradation by inactivating Rab8A and Rab10, and not Rab29. However, in their system Rab29 can rescue the endo-lysosomal deficits caused by mutant LRRK2 and inactive Rab8A and Rab10.

To confirm that these findings are true in the context Parkinson's disease - it may be worth looking at dopaminergic neurons from IPSCs or a dopaminergic cell line that contains dopamine. Although, this may be beyond the scope of the current manuscript, but it would be very insightful to know whether these findings also hold true for in dopaminergic neurons? Does mutant LRRK2 impair endolysosomal degradation in a Rab8 and Rab10 dependent manner in dopaminergic neurons? I recognize that is may be difficult to achieve, if the authors can't confirm that these proteins function similarly in a more PD-relevant model. The authors should mention address this limitation in the discussion. It is possible that a Hela cell will function very differently then a post-mitotic dopaminergic neuron. 

Are these trafficking and clearance observations specific for EGF? From a PD perspective, it would be interesting to see how alpha synuclein impairs trafficking and clearance through the endolysosomal pathway? If the cells were treated with oligomeric alpha-synuclein, could the synuclein activate LRRK2 and cause similar endolysosomal deficits similar to those observed with the LRRK2 mutation?

Minor comment - The authors state that the G2019S mutation delayed the clearance of EGF, however, based on the data in Figure 1, it looks as though the G2019S impaired clearance. Unless, the amount of intracellular EGF eventually looks statistically similar to pCMV control cells.   

Author Response

We thank the referee for his/her constructive input.

We agree that it will be important in the future to look at trafficking deficits in dopaminergic neurons from IPSCs or dopaminergic cell lines, even though we strongly believe that this is beyond the scope of the present manuscript. In addition, whilst PD involves dopaminergic cellular demise, it is currently unknown whether these changes occur in a cell-autonomous manner, such that it will also be important to look at endolysosomal trafficking deficits in other potentially disease-relevant cell types. However, and as requested by the referee, we now state in the revised manuscript (discussion):

".....and in the future, it will be important to determine the combined levels of both phospho-proteins, and whether LRRK2-mediated endolysosomal trafficking alterations occur in a manner dependent on RAB8A and RAB10 in dopaminergic neurons or other disease-relevant cell types."

The referee enquires whether the trafficking deficits are specific to the EGFR. They are not, as deficits in the endocytosis/internalization have also been observed with respect to the dopamine D1 receptor in cell lines as well as intact brain (Rassu et al., PLoS One, 2017), and with respect to the transferrin receptor in cell lines (Heaton et al., Neurobiol. Dis., 2020). We now mention these findings in the discussion of the revised manuscript, and have added the relevant references:

"In addition, endocytic trafficking deficits have also been described for the dopamine D1 receptor and the transferrin receptor (Rassu et al., PLoS One, 2017; Heaton et al., Neurobiol. Dis., 2020), and it will be interesting to determine whether they are due to the same LRRK2-mediated inactivation of RAB8A and RAB10 as described here."

The referee enquires whether oligomeric alpha-synuclein may activate LRRK2 to cause endolysosomal trafficking deficits similar to those observed with pathogenic LRRK2 expression. We have not probed for this possibility in the present manuscript since the link between pathogenic LRRK2 and alpha-synuclein remains controversial (eg most patients with LRRK2 mutations do not display Lewy bodies in postmortem brain (Henderson et al., Acta Neuropathol. Commun., 2019)), and alpha-synuclein fibrils tend to be rather cytotoxic per se. However, it will be interesting in future to co-express alpha-synuclein along with LRRK2 and determine whether this enhances the LRRK2-mediated trafficking deficits, along the lines mentioned by the referee.

Minor: we stated "delayed clearance" rather than "impaired clearance". However, the referee is correct, and since we have not analyzed the remaining fluorescent EGF beyond the 30 min timepoint, the correct statement here is "impaired". We have changed this statement in the manuscript accordingly.

Reviewer 2 Report

Rab proteins have been known as essential regulator of intracellular vesicle trafficking. Therefore, defining underlying molecular mechanisms of each Rab proteins in specific intracellular functional processes is of interest to researchers in various fields. Through recent studies on large-scale proteomics study and in vitro phosphorylation profiling, a subset of Rab proteins was identified as authentic substrates of LRRK2, implying Rab proteins-mediated regulation on physiological and pathological roles of LRRK2. Previous reports and here, the authors suggested a subset of Rab proteins are involved in LRRK2 linked deficit in endolysosomal trafficking/degradation of EGFR. The manuscript presented by authors is well conducted and very informative with a convincing set of experimental data. To improve your manuscript, several comments are listed below.

  1. Does LRRK2 G2019S and/or Rab10 knock down-induced trafficking deficit of EGFR have effect on EGFR downstream signaling?
  2. Rab10 is also known as LRRK2 kinase substrate, Have you tested the effect of phosphor-deficit and mimic mutants on EGF binding and degradation?
  3. Rab10 active form (Q68L) rescues LRRK2 G2019S mediated deficit in EGF binding and degradation, but not Rab10 wildtype. But Rab29 wildtype rescues LRRK2 G2019S mediated deficit in EGF binding and degradation, but not Rab29 active form (Q67L). Could you explain the different effect of Rab10 and Rab29?
  4. In both previous report (Rivero-Rios et al. 2019) and here, the authors suggested a subset of Rab protein (Rab7, 8, 10, 11, 18, 29 etc) are involved in LRRK2 mediated EGFR recycling and degradation through endolysosomal pathways. In my opinion, summarizing the roles and positioning of each Rab protein in these regulation will be helpful to the readers.

Author Response

We thank the referee for his/her constructive input. We have addressed comments as outlined below:

  1. Internalization of the activated EGFR allows for signaling from intracellular sites, and endocytosis of the EGFR for example is required for the full activation of MAP kinase signaling cascades (eg Vieira et al., Science, 1996). Therefore, and given the observed trafficking deficits described here, one may indeed expect alterations in EGFR downstream signaling in the context of pathogenic LRRK2 expression. However, we have not probed for such alterations, but merely used the EGFR as a model receptor to study LRRK2-mediated trafficking alterations, since there is no published link between EGFR trafficking and PD (even though there are clear links between MAP kinase signaling alterations in the context of pathogenic LRRK2).

  1. We have not employed phospho-deficient and phospho-mimetic Rab10 mutants, since previous studies established that these mutants are non-functional. Rab10-TA (phospho-deficient) is not membrane-localized, and Rab10-TE (phospho-mimetic) shows membrane localization distinct from wildtype Rab10 and is unable to interact with RILPL1 (in contrast to endogenously phosphorylated Rab10), and therefore is not mimicking the phosphorylated state of Rab10 (Dhekne et al., Elife, 2018).

  1. Please note that expression of wildtype Rab proteins is often without effect, since activation of the Rabs depends on GEFs (GDP/GTP exchange factors) which are rate-limiting in most cases. This is why effects are observed with the active form of Rab10 (Q68L), which mimicks the GTP-locked form of the Rab protein, but not with wildtype Rab10. Rab29 is a peculiar Rab protein. Previous studies have shown that the Q67L mutation which in other Rab proteins is expected to lock the protein in an active conformation is not localized to the Golgi, and binds Rab29 more weakly than wildtype Rab29 (eg. Beilina et al., PNAS, 2014). Therefore, such canonical mutation cannot be used to study Rab29, as not mimicking a GTP-locked form of the protein. We refer to this in the manuscript (line 259): " In contrast, two mutant Rab29 versions previously described to cause Rab29 inactivation...."

  1. In our previous report, Rab18 was merely used as a negative control to show that Rab protein expression per se does not cause a rescue of the LRRK2-mediated trafficking deficits. We feel that we extensively discuss the impact of pathogenic LRRK2, or knockdown of Rab8 or Rab10 on causing a decrease in active Rab7, with downstream effects on endolysosomal trafficking, even though the mechanism underlying such decreased Rab7 activity remains unknown. Importantly, we also discuss the Rab29-mediated rescue of the trafficking deficits mediated by pathogenic LRRK2, and provide a model suggesting that this may occur via positive regulation of Rab8/10 (Figure 8), even though this will require further experimental validation. In our previous report, we showed that Rab11 and Rabin8 can rescue the LRRK2-mediated EGFR trafficking deficits. Rabin8 serves as a GEF for Rab8a, and in turn is activated by Rab11. Therefore, the link to rab11 only relates to regulation of Rab8 activity. Therefore, and for simplicity, we would prefer not to further discuss these previously published data in the present manuscript.